# Four New Endophytic *Apiospora* Species Isolated from Three *Dicranopteris* Species in Guizhou, China

**DOI:** 10.3390/jof9111096

**Published:** 2023-11-10

**Authors:** Jing-Yi Zhang, Meng-Lan Chen, Saranyaphat Boonmee, Yu-Xuan Wang, Yong-Zhong Lu

**Affiliations:** 1School of Food and Pharmaceutical Engineering, Guizhou Institute of Technology, Guiyang 550003, China; zjingyi127@gmail.com (J.-Y.Z.); cml163abc@163.com (M.-L.C.); 13375985192@163.com (Y.-X.W.); 2School of Science, Mae Fah Luang University, Chiang Rai 57100, Thailand; saranyaphat.boo@mfu.ac.th; 3Center of Excellence in Fungal Research, Mae Fah Luang University, Chiang Rai 57100, Thailand

**Keywords:** seven taxa, asexual morph, dematiaceous conidia, medicinal ferns, phylogeny, taxonomy

## Abstract

Endophytic fungi isolated from medicinal ferns serve as significant natural resources for drug precursors or bioactive metabolites. During our survey on the diversity of endophytic fungi from *Dicranopteris* species (a genus of medicinal ferns) in Guizhou, *Apoiospora* was observed as a dominant fungal group. In this study, seven *Apiospora* strains, representing four new species, were obtained from the healthy plant tissues of three *Dicranopteris* species—*D*. *ampla*, *D*. *linearis,* and *D*. *pedata*. The four new species, namely *Apiospora aseptata*, *A*. *dematiacea*, *A*. *dicranopteridis,* and *A*. *globosa*, were described in detail with color photographs and subjected to phylogenetic analyses using combined LSU, ITS, TEF1-α, and TUB2 sequence data. This study also documented three new hosts for *Apiospora* species.

## 1. Introduction

Apiosporaceae, typified by *Apiospora* Sacc., was established by Hyde et al. [1] to accommodate *Arthrinium*-like taxa characterized by a basauxic, *arthrinium*-like conidiogenesis producing apiospores [2,3]. Currently, only three genera, *Apiospora*, *Arthrinium* Kunze, and *Nigrospora* Zimm, are accepted in this family [4,5]. *Apiospora* Sacc. was introduced by Saccardo, P. A. [6] within the family Apiosporaceae (Amphisphaeriales, Sordariomycetes), with *A*. *montagnei* Sacc. as the type species. Due to the morphological similarities between the genera *Apiospora* and *Arthrinium*, they were long considered synonymous based on the one fungus one name principle [7,8,9,10,11]. It was not until the study by Pintos, Á. & P. Alvarado [12] that they were clarified as separate genera based on genetic, ecological, and morphological evidence. This delineation was confirmed and supported by subsequent studies [13,14,15,16,17,18,19,20]. Evidence of genome draft was employed within the fungal group *Arthrinium*/*Apiospora* for the first time, which also supported them into two separate genera [21]. Currently, 157 epithets under the genus *Apiospora* are listed in Index Fungorum (September 2023), while 116 epithets are listed in Species Fungorum. Out of these, molecular data have confirmed 91 *Apiospora* species [12,13,14,15,17,18,19,20,22].

The sexual morph of *Apiospora* is characterized by immersed to erumpent, multi-loculate, perithecial ascostromata, unitunicate, broadly clavate to cylindric–clavate asci, and hyaline, ellipsoidal, inequilaterally 2-celled ascospores with or without a gelatinous sheath [5,9,10,14,22,23]. The asexual morphs of *Apiospora* include hyphomycetes and coelomycetes. The hyphomycetous asexual morphs feature septate, subhyaline, or brown conidiophores emerging from basal cells or reduced to conidiogenous cells, basauxic conidiogenous cells, and typically globose to subglobose, aseptate conidia that appear lenticular or obovoid from side view [11,13,16,18,19,24]. The coelomycetous asexual morph of *Apiospora* is marked by its erumpent, pustulate, coriaceous conidiomata, hyphoid conidiophores, blastic, integrated, determinate, doliiform or cylindrical conidiogenous cells, and oval, brown conidia, which may have a truncate basal scar and a germ slit [2,9,20].

Endophytes are endosymbiotic flora, microorganisms that colonize the internal tissues of healthy plants without causing any direct, noticeable negative effects [25,26]. The taxonomic research of endophytic fungi has become a popular trend not only because of their beneficial effects on plants but also because of compounds including antibiotics and other compounds of therapeutic significance [26,27,28]. That is, endophytic fungi (especially medicinal plants) possess significant potential to discover or synthesize more bioactive compounds and mimic the structure and function of host compounds [25,28,29,30], which shows a new source of potentially useful pharmaceutical compounds [26,28,31].

*Dicranopteris* Bernh. is an ancient and widespread fern genus belonging to the family Gleicheniaceae (Filicopsida) found in tropical and subtropical ecosystems [32,33]. It belongs to a group of medically important ferns known for their significant pharmacological effects, including removing blood stasis, clearing heat and diuresis, anticancer, antinociception, and anti-inflammation [34,35,36,37,38]. Extracts from *Dicranopteris* are rich in bioactive compounds and have the potential to yield new structural compounds [34,39].

In this study, we examined endophytic fungi isolated from three *Dicranopteris* species (*D*. *ampla*, *D*. *linearis,* and *D*. *pedate*) in Guizhou, China, aiming to explore the diversity of fungi with research significance. We isolated nearly a thousand endophytic taxa from various parts of the three *Dicranopteris* species, belonging to 146 genera, based on NCBI searches of the ITS and LSU sequence data. Among these isolates, *Apiospora* emerged as a common genus. Within this collection, seven taxa were identified herein as four endophytic *Apiospora* species new to science, viz. *A*. *aseptata*, *A*. *dematiacea*, *A*. *dicranopteridis,* and *A*. *globosa*. To further determine the taxonomic placement of these four *Apiospora* species, we employed phylogenetic analyses using combined LSU, ITS, TEF1-α, and TUB2 sequence data, complemented by morphological features. A backbone tree of Apiosporaceae is provided in this study.

## 2. Materials and Methods

### 2.1. Collection and Isolation

Fresh, healthy plant tissues (leaves, rhizomes, roots, and stems) from three *Dicranopteris* species were collected along with relevant metadata (date, habitat, and locality). Samples were transported to the laboratory and processed for fungal isolation within 48 h. Healthy tissue pieces were first washed under running tap water. Surface sterilization of plant tissues followed the method described by Nontachaiyapoom et al. [40], with some modifications. To eliminate epiphytic microorganisms, the materials were surface-sterilized on a benchtop by immersing them in 75% (*v*/*v*) ethanol for 1–3 min (ca. 1 min for leaves and stems; ca. 3 min for rhizomes and roots). They were then rinsed with sterilized distilled water for 2 min, followed by a soak in 10% (*v*/*v*) NaClO for 0.5–2 min (ca. 0.5 min for leaves and stems; ca. 2 min for rhizomes and roots). The tissues were then rinsed with sterile distilled water three times in succession. After drying the sterilized plant tissues on sterilized filter paper, they were cut into approximately 2 mm^2^ pieces using a sterile blade. These small pieces were placed on fresh potato dextrose agar (PDA) containing antibiotics (50 μg/mL penicillin) and cultivated at 25 °C. Once fungal hyphae growth was observed emerging from the plant segments, the hyphae were picked from the edge of the colonies and transferred to fresh PDA media to obtain the pure cultures.

### 2.2. Morphological Study and Conservation

Isolates were grown on a PDA for one week, and cultural characteristics such as size, shape, color, and texture were recorded. These characteristics were further examined using a stereomicroscope (SMZ168-BL, Motic, Shanghai, China). Micro-morphological characteristics were described based on cultures that sporulated on either water agar (WA) or PDA [11,16,19,23,41,42]. These were photographed using an ECLIPSE Ni-U compound microscope (Nikon, Tokyo, Japan) equipped with an EOS 90D digital camera (Canon, Tokyo, Japan). Measurements of conidiophores, conidiogenous cells, conidia, and mycelia were conducted using the Tarosoft (R) Image Frame Work (version 0.9.7). Figures and the photoplates were processed with Adobe Illustrator CS6 v. 24.0.1 (Adobe Systems, San Jose, CA, USA). Dried materials were deposited in the Herbarium of Cryptogams, Kunming Institute of Botany Academia Sinica (HKAS), Kunming, China, and the Herbarium of Guizhou Academy of Agricultural Sciences (GZAAS), Guiyang, China. Living cultures were deposited at the Kunming Institute of Botany, the Chinese Academy of Sciences (KUNCC), and the Guizhou Culture Collection (GZCC). Faces of Fungi and Index Fungorum numbers were registered in accordance with the guidelines presented in Jayasiri et al. [43] and Index Fungorum (http://www.indexfungorum.org/Names/Names.asp; accessed on 15 September 2023).

### 2.3. DNA Extraction, PCR Amplification and Sequencing

Genomic DNA was extracted from fresh fungal mycelia using the Biospin Fungus Genomic DNA Extraction Kit (BioFlux^®^, Shanghai, China) according to the manufacturer’s instructions. Four primer pairs, namely LR0R and LR5 [44], ITS5 and ITS4 [45], EF2 and EF1-728f [46,47], and T1 and Bt2b [48,49], were employed to amplify the large subunit of the ribosomal DNA (LSU), the internal transcribed spacer (ITS), the elongation factor 1-alpha (TEF1-α), and the β-tubulin (TUB2) gene regions, respectively. The polymerase chain reaction (PCR) was carried out in a 50 μL reaction volume containing 2 μL of DNA template, 2 μL of each forward and reverse primer (10 μM), 25 μL of 2× Taq PCR Master Mix with blue dye (Sangon Biotech, China), and 19 μL of distilled–deionized water. The amplification conditions for LSU, ITS, TEF-1α, and TUB2 were based on the protocol described by Feng et al. [23]. Successful PCR products were sent to Sangon Biotech (Shanghai, China) for purification and sequencing. The sequences generated in this study have been deposited in NCBI GenBank (Table 1).

### 2.4. Alignments and Phylogenetic Analyses

The quality of the original sequences was checked using BioEdit v. 7.1.3.0 [50] and assembled with SeqMan v. 7.0.0 (DNASTAR, Madison, WI, USA). Consensus sequences underwent BLASTn analysis in the NCBI GenBank database for preliminary identification of similar sequences. Taxa (Table 2), including type and additional strains of *Apiospora* species and related genera (*Nigrospora* and *Arthrinium*) in Apiosporaceae, were selected for the phylogenetic analyses based on data obtained from Genbank and previous studies [8,11,12,14,19,20,23,41,51]. Sequence alignment was performed using MAFFT v.7.0 (https://mafft.cbrc.jp/alignment/server/; accessed on 20 August 2023) [52] and subsequently manually verified in BioEdit 7.1.3.0 [50]. The phylogenetic relationships, based on a combined LSU–ITS–TEF-1α–TUB2 dataset, were analyzed using both maximum likelihood (ML) and Bayesian inference (BI) criteria.

Maximum likelihood (ML) analysis was conducted on the CIPRES web portal (https://www.phylo.org/portal2/home.action; accessed on 20 August 2023) using the RAxML-HPC Blackbox (8.2.10) tool with rapid bootstrap analysis and 1000 bootstrap replicates [53,54]. The final tree was selected from the suboptimal trees of each run by comparing likelihood scores under the GTRGAMMA substitution model.

Posterior probabilities (PP) [55] were calculated using the Bayesian Markov Chain Monte Carlo (BMCMC) sampling method in MrBayes 3.2.7a via CIPRES [53]. The appropriate substitution model best fitting the DNA evolution model for the combined dataset was determined using MrModeltest v.2.3 [56]. For the LSU, ITS, and TUB2 datasets, GTR+I+G was selected, whereas HKY+I+G was selected for TEF1-α. Four simultaneous Markov chains run for 1 million generations, with trees sampled every 100 generations, yielding 10,000 trees. The first 2000 trees, representing the burn-in phase, were discarded, and the remaining 8000 trees were used for calculating posterior probabilities (PP) in the majority rule consensus tree [57].

Phylogenetic trees were visualized using FigTree v. 1.4.4 [58] and adjusted using Adobe Illustrator CS6 (Adobe Systems, San Jose, CA, USA). 

## 3. Results

### 3.1. Phylogenetic Analysis

Seven endophytic taxa with asexual morphs, isolated from three *Dicranopteris* species, were identified as *A. aseptata*, *A*. *dematiacea*, *A*. *dicranopteridis,* and *A*. *globosa* spp. nov. within the genus *Apiospora* (Apiosporaceae, Amphisphaeriales) (Table 1). The combined LSU (840 bp), ITS (623 bp), TEF1-α (534 bp), and TUB2 (207 bp) sequence alignment comprised 135 taxa, with *Seiridium phylicae* (CPC 19962 and CPC 19965) serving as the outgroup taxa.

The dataset contained 2204 characters after alignment. The matrix presented 1181 distinct alignment patterns, with 25.29% being completely undetermined characters or gaps. Base frequencies and rates were A = 0.235259, C = 0.247975, G = 0.262058, and T = 0.254709; substitution rates were AC = 1.153887, AG = 2.614757, AT = 1.016680, CG = 0.924945, CT = 4.569867, and GT = 1.000000, with a tree length of 4.582782. The distribution shape parameter α equaled 0.227323. The tree topologies generated from both RAxML and Bayesian analyses were similar, showing no significant conflicts. The best-scoring RAxML tree is shown in Figure 1, with a final likelihood value of −25,698.717456. This phylogenetic tree revealed that the new species *Apiospora aseptata* (KUNCC 23-14169) clusters with two unidentified *Apiospora* taxa (SAUCC 1429 and SAUCC 1430), albeit with weak support. *Apiospora dematiacea* is closely related to the species, which includes four taxa of *Apiospora hydei* (LC7105, LC7103, CBS 114990, and SICAUCC 22-0032). Four isolates representing the new species, *Apiospor dicranopteris* (GZCC 23-0708, GZCC 23-0712, KUNCC 23-14177, and KUNCC 23-14171), form a distinct clade, which is basal to *A*. *koreana* (KUC21332) and *A*. *qinlingensis* (CFCC 52303). Lastly, *Apiospor globosa* forms its own clade, being a sister to *A*. *neosubglobosa* (KUMCC 16-0203 and JHB 006).

### 3.2. Taxonomy

*Apiospora aseptata*, J.Y. Zhang and Y.Z. Lu, sp. nov. (Figure 2)

Index Fungorum number: IF901115; Facesoffungi number: FoF14873.

Etymology: referring to the aseptate conidia.

Culture characteristics: Colonies on PDA are medium circular, spread, flat with an entire edge, with thin aerial hyphae, reaching ca. 50 mm diam after 10 d at 25 °C, grey-brown from above, yellow–brown in reverse. Mycelium consists of septate, branched, hyaline to brown hyphae.

Description: Endophytic in the healthy roots of *Dicranopteris pedata*. Sexual morph: Undetermined. Asexual morph: *Conidiophores* are cylindrical, septate, branched, rough-walled, flexuous, and often reduced to conidiogenous cells. *Conidiogenous cells* ca. 3.5 µm wide, aggregated in clusters on hyphae, solitary, mono-polyblastic, cylindrical to subglobose, hyaline to brown. *Conidia* amerospores, aseptate, globose or sub globose, 7–9.5 (–13) µm diam. ($\bar{x}$ = 8 µm, n = 40) in surface view, subglobose; 6–8.5 × 5–7 µm ($\bar{x}$ = 7 × 6 µm, n = 20) from side view, lenticular with a pale longitudinal germ slit, smooth to finely roughened, occasionally micro guttules, pale brown to brown.

Material examined: China, Guizhou Province, Qianxinan Buyi, and Miao Autonomous Prefecture, Chengheng County, isolated from the healthy leaf of *Dicranopteris pedata* near the roadside, 16 March 2022, J.Y. Zhang, 138-3 (HKAS 129875, holotype); ex-type living cultures, KUNCC 23-14169.

GenBank accession numbers: (LSU) OR590335, (ITS) OR590341, (TEF1-α) OR634949, and (TUB) OR634943.

Notes: *Apiospora aseptata* aligns well with the characteristics of the genus *Apiospora* and is most similar to *A*. *pseudoparenchymaticum* in the shape of its conidiogenous cells and conidia. However, they differ in conidial size [11]. *Apiospora aseptata* has noticeably smaller conidia than *A*. *pseudoparenchymaticum* (7–9.5 (–13) µm diam. vs. 13.5–27 × 12–23.5 µm). In phylogenetic analysis, *Apiospora aseptata* clusters with *Apiospora* sp. strains (SAUCC 1429 and SAUCC 1430) and forms a sister relationship with the clade that includes *A*. *arctoscopi* (KUC21331), *A*. *jiangxiensis* (LC 4577), and *A*. *obovata* (LC 4940). Unfortunately, we could not compare the morphological characteristics of the *Apiospora* sp. strains (SAUCC 1429 and SAUCC 1430), as the morphology of these strains has not been reported. Based on both phylogeny and morphology, we introduce *Apiospora aseptate* as a new species.

*Apiospora dematiacea* J. Y. Zhang and Y. Z. Lu, sp. Nov. (Figure 3)

Index Fungorum number: IF901116; Facesoffungi number: FoF14874.

Holotype: HKAS 129910

Etymology: referring to its dematiaceous spore.

Culture characteristics: *Colonies* on PDA medium circular, cottony, edge entire, flat, spreading, with abundant aerial mycelia, zonate with one concentric circle, reaching 47 mm diam after 10 d at 25 °C, white from above, yellowish white to grey to light yellow from center to edge in reverse. *Vegetative hypha* septate, branched, hyaline to light brown.

Description: Endophytic in the stems of *Dicranopteris ampla*. Sexual morph: Undetermined. Asexual morph: *Conidiophores* reduced to conidiogenous cells, cylindrical, septate, hyaline. *Conidiogenous cells* are cylindrical to subglobose, aggregated in clusters on hyphae, smooth, hyphae-like, and hyaline to brown. Conidia aseptate, globose to ellipsoid in surface view, 14.5–18(–20) µm diam. ($\bar{x}$ = 16.5 µm, n = 30), lenticular to lageniform from side view, 18.5–23(–25) × 10–13 µm diam. ($\bar{x}$ = 21.5 × 11.5 µm, n = 30), with longitudinal, pale germ slit. *Sterile cells* up to 32 µm long, 9–12(–16) µm ($\bar{x}$ = 10.5 µm, n = 20) wide, elongated, mixed among conidia brown, rarely truncate, and have a darkened scar at the base.

Material examined: CHINA, Guizhou Province, Qianxinan Buyi and Miao Autonomous Prefecture, Ceheng County (24°59′44″ N 105°50′16″ E), isolated from the healthy stem of *Dicranopteris ampla* near the roadside, 16 March 2022, J.Y. Zhang, 307-1 (HKAS 129910, Holotype), ex-living cultures, KUNCC 23-14202.

GenBank accession numbers: (LSU) OR590339, (ITS) OR590346, (TEF1-α) OR634953, and (TUB) OR634948.

Note: *Apiospora dematiacea* morphologically resembles *A*. *hydei*, characterized by conidiogenous cells aggregated in clusters on hyphae and globose conidia in surface view, with a pale equatorial slit when viewed from the side [8]. However, *A*. *dematiacea* is distinguishable from *A*. *hydei* due to its hyphae-like conidiogenous cells and more varied conidial shapes that include sterile cells. From a phylogenetic perspective, while *Apiospora dematiacea* shares a sister relationship with *A*. *hydei*, it constitutes a distinct lineage. A comparison of nucleotide base pairs between the ex-type strain of *A*. *hydei* (CBS 114990) and our newly isolated strain of *Apiospora dematiacea* (KUNCC 23-14202) reveals differences of 1183/1185 bp (99%), 578/579 (99%, including 1 gap), 411/432 bp (95%, including 12 bp gaps), and 778/794 bp (98%, including 2 bp gaps) in the LSU, ITS, TEF1-α, and TUB2 sequences, respectively. This confirms that they are distinct species.

*Apiospora dicranopteridis* J.Y. Zhang and Y.Z. Lu, sp. nov. (Figure 4)

Index Fungorum number: IF901117; Facesoffungi number: FoF14875

Etymology: referring to the fungal host genus, *Dicranopteris.*

Holotype: HKAS 129877

Culture characteristics: Colonies on PDA medium circular, edge entire, floccose at the surface with dense, white aerial mycelia, growly fast, reaching 55 mm diam after 10 d at 25 °C, cottony, velvety, loose, white from above, yellow to yellowish whites in reverse. Vegetative hypha: septate, branched, sometimes coiled, guttulate, hyaline to pale brown.

Description: Endophytic in the stems of *Dicranopteris pedata*. Sexual morph: Undetermined. Asexual morph: *Conidiophores* are cylindrical, septate, branched, smooth-walled, and often reduced to conidiogenous cells. *Conidiogenous cells* 6–15 × 3.5–10 µm ($\bar{x}$ = 10 × 6 µm, n = 25) µm, solitary to aggregated in clusters arising from dense aerial hyphae, mono- to polybasic, sympodial, sub-globose to doliiform to cylindrical, smooth, subhyaline. *Conidia* amerospores, aseptate, globose or sub globose, 10.5–13 µm diam. ($\bar{x}$ = 11.5 µm, n = 15), cylindrical to broadly clavate 14–17(–22) × (6–)8–10.5 µm ($\bar{x}$ = 16 × 9 µm, n = 8), with rounded at the apex and a slightly narrower and truncate base, smooth to finely roughened guttules, without an equatorial germ slit, hyaline to pale brown.

Material examined: China, Guizhou Province, Qianxinan Buyi and Miao Autonomous Prefecture, Ceheng County (24°59′44″ N 105°50′16″ E), isolated from the healthy stems of *Dicranopteris pedata* nearby the roadside, 16 March 2022, J.Y. Zhang, 139-2 (HKAS 129877, holotype), ex-type living cultures, KUNCC23-14171; Ibid., isolated from the root of *D*. *pedate*, 16 March 2022, J.Y. Zhang, 170-4 (GZAAS 23-0780, paratype), living cultures, KUNCC 23-14177; Ibid, Anlong County, Jia Jia Ya Kou (24°59′23″ N; 105°35′20″ E), isolated from the healthy leaf of *D*. *pedate*, 16 March 2022, J.Y. Zhang, 223-4 (HKAS 129895, paratype), living cultures, GZCC 23-0712; Ibid, isolated from the healthy rhizome of *D*. *ampla*, 16 March 2022, J.Y. Zhang, 225-1 (HKAS 129898, paratype), living cultures, GZCC 23-0708.

GenBank accession numbers: KUNCC23-14171: (LSU) OR590336, (ITS) OR590342, (TEF1-a) OR634950, (TUB2) OR634944; KUNCC 23-14177: (LSU) OR590337, (ITS) OR590343, (TEF1-a) OR634951, (TUB2) OR634945; GZCC 23-0712: (LSU) OR590338, (ITS) OR590345, (TEF1-a) OR634952, (TUB2) OR634947; GZCC 23-0708: (ITS) OR590344, (TUB2) OR634946.

Notes: *Apiospora dicranopteridis* is morphologically distinct from other *Apiospora* species by its mono- or polyblastic, elongated cylindrical conidiogenous cells and globose to cylindrical to broadly clavate, hyaline to pale brown conidia. Phylogenetically, four strains (GZCC 23-0708, GZCC 23-0712, KUNCC23-14177, and KUNCC23-14171) representing *Apiospora dicranopteridis* sp. nov. formed a distinct clade. They share a sister relationship with *A*. *koreana* (KUC21332) and *A*. *qinlingensis* (CFCC 52303), reinforcing the notion that they are separate species.

*Apiospora globosa* J.Y. Zhang and Y.Z. Lu, sp. nov. (Figure 5)

Index Fungorum number: IF901402; Facesoffungi number: FoF14658.

Etymology: referring to the globose to subglobose conidia

Holotype: HKAS 129921

Culture characteristics: *Colonies* on WA medium irregulate, with several dark spots, flat with undulate edge, reaching 33 mm diam after 15 d at 25 °C, hyaline to light brown. *Vegetative hyphae are* thin, sparse, septate, branched, guttulate, and hyaline, some curled in a ring structure.

Description: Endophytic in the stems of *Dicranopteris linearis*. Sexual morph: Undetermined. Asexual morph: *Conidiophores* undistinguishable, hyphae-like. *Conidiogenous cells* are undistinguishable and hyphae-like. *Conidia* produced directly from vegetative hypha inside the WA culture, 4.5–8.5 µm diam ($\bar{x}$ = 6 µm, n = 20), aseptate, globose to subglobose, smooth to finely roughened, light yellow to gold to black.

Material examined: China, Guizhou Province, Anshun City, Ziyun Miao Buyi Autonomous County, Getu River Scenic Spot (25°48′26″ N 106°4′24″ E), isolated from the healthy stem of *Dicranopteris linearis* in a disturbed forest, 2 August 2022, J.Y. Zhang, S4-1 (dry WA culture, HKAS 129921, holotype; dry culture of WA-carrot mixture, GZAAS 23-0790), ex-type living cultures, KUNCC 23-14210.

GenBank accession numbers: (LSU) OR590340, (ITS) OR590347, and (TEF1-α) OR634954.

Notes: Phylogenetically, *Apiospora globosa* forms a distinct clade that is sister to the species *Apiospora neosubglobosa* (KUMCC 16-0203 and JHB 006). While *Apiospora neosubglobosa* has been described with only a sexual morph [9,22], our new species produces an asexual morph in culture. Morphologically, *Apiospora globosa* resembles *A*. *xenocordella* in conidial shape but has notably different conidiogenous cells and conidial size (4.5–8.5 µm diam. vs. 9–10 µm diam.). *Apiospora globosa* possesses indistinct, hyphae-like conidiogenous cells, whereas *A*. *xenocordella* features globose to clavate to doliiform conidiogenous cells [8].

## 4. Discussion

The genus *Apiospora* is relatively well studied, with species distributed across tropical, subtropical, temperate, and cold climates globally [8,12,14,41,51]. Members of the *Apiospora* species can function as endophytes [16,19,59,60], pathogens [8,19,61], or saprobes [5,10,13,23], found on various hosts, including various plants, air, water, soil debris, home dust, food, and the gut of insects [8,11,12,14,59,61]. They do not exhibit a clear lifestyle preference or pronounced sensitivity to environmental change. A fungus-host distribution of *Arthrinium* species (most of which have been synonymized under *Apiospora*) was provided by Wang et al. [11]. The data showed that Poaceae and Cyperaceae are the dominant host plant families, especially the former [10,12,14,17,20,22,42,62,63]. There have been no previous reports of *Apiospora* species from ferns, likely due to the neglect of fungi on ferns [64,65]. In this study, four new species (*Apiospora aseptata*, *A*. *dematiacea*, *A*. *dicranopteridis,* and *A*. *globosa*) were reported as endophytes isolated from three medicinal ferns—*Dicranopteris ampla*, *D*. *linearis*, and *D*. *pedate*—based on evidence from morphology and phylogenetic analyses of a concatenated dataset of LSU, ITS, TEF1-α, and TUB2 sequences. This study represents the first report of *Apiospora* species from *Dicranopteris* species, expanding the host diversity knowledge of *Apiospora* species.

The *Apiospora*-*Arthrinium* group has made certain achievements in bioactive secondary metabolites, with a high interest in agriculture, food, and the pharmaceutical industry [66,67,68,69]. The evaluation of the biological activities of *Apiospora*-*Arthrinium* spp. revealed this group has relatively biological activities for antifungal, antioxidant, and cellulolytic activity, especially *Apiospora saccharicola* [66]. A quick guide to secondary metabolites from the *Apiospora*–*Arthrinium* group was provided by Overgaard et al. [69], including the knowledge of 269 secondary metabolites and emphasizing some of the known biological or toxic compounds. For example, several species, including *Apiospora arundinis*, *A*. *aurea*, *A*. *phaeosperma*, *A*. *sacchari*, *A*. *saccharicola*, *A*. *serenensis*, *A*. *terminalis*, and *Apiospora*/*Arthrinium* spp., can produce 3-nitropropionic acid, which is associated with the food safety problem of poisonings or even deaths [70,71,72]. The study results of the genome sequence provided by Sørensen et al. [21] revealed that the *Apiospora*-*Arthrinium* group holds a high number of secondary metabolite gene clusters, which has attracted more attention to the compounds of this group [21,69]. In the current research context, we newly obtained seven endophytic taxa from the medicinal ferns of *Dicranopteris* rich in bioactive compounds and identified them as four new species (*A*. *aseptata*, *A*. *dematiacea*, *A*. *dicranopteridis,* and *A*. *globosa*) in the genus *Apiospora*, a fungal group that is currently attracting attention. These isolates are valuable fungi that are expected to explore secondary metabolites, which will also be the future research direction of our research.

All strains were initially cultured on a PDA medium. Most *Apiospora* strains sporulate on PDA substrate naturally under conventional conditions (room temperature 25–28°, natural light), which is consistent with many studies [8,11,16,19,23,41]. However, *Apiospora globosa* (KUNCC 23-14210) failed to sporulate on the PDA medium. Sporulation was later induced on various media, including CMM (Corn Meal Medium), MEA (malt extract agar), OA (oatmeal agar), SNA (synthrtic nutrient-poor agar), and WA (water agar). Eventually, sporulation of *Apiospora globosa* (KUNCC 23-14210) was successfully induced on WA and a WA-carrot mixture. During investigations of endophytic fungal diversity isolated from *Dicranopteris* in Guizhou, China, hundreds of isolates were selected from nearly a thousand endophytic strains for sporulation induction across multiple media (CMM, MEA, OA, PDA, SNA, and WA). Results showed that the sporulation rate on the WA medium exceeded that of other media. Therefore, WA is prioritized for inducing sporulation in plant endophytic fungi when the sporulation mechanism for this fungal group is not documented in previous publications.

## Figures and Tables

**Figure 1 jof-09-01096-f001:**
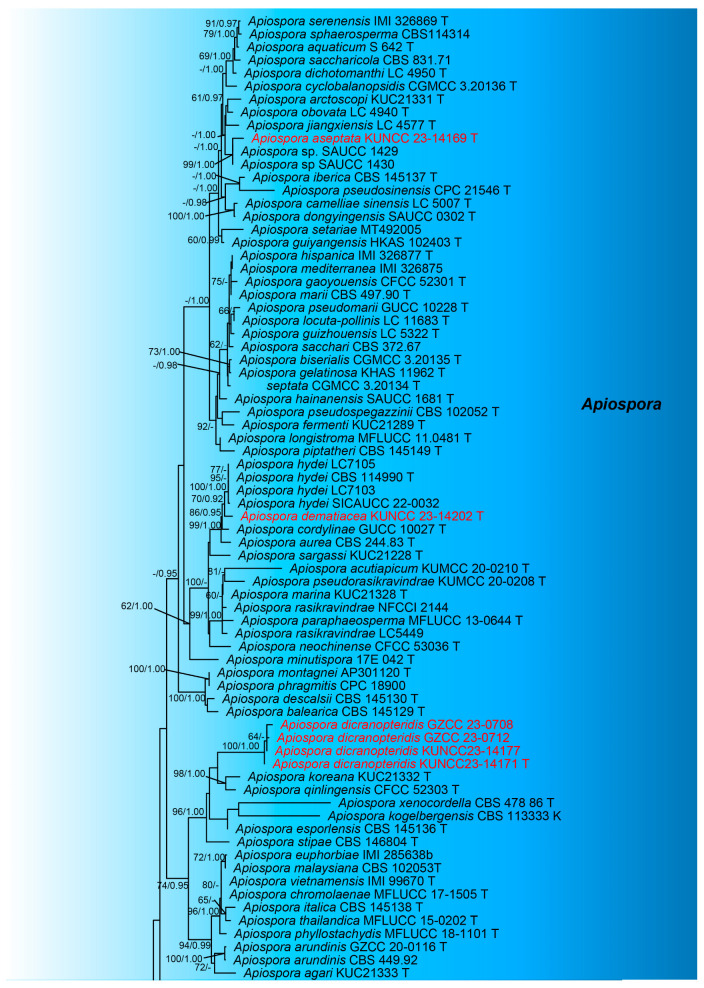
The phylogenetic tree generated from ML analysis is based on a concatenated LSU–ITS–TEF1α–TUB2 dataset for the family Apiosporaceae. Bootstrap support values for ML greater than 60% and Bayesian posterior probabilities (PPs) greater than 0.95 were indicated above or below the nodes as ML/PP. *Seiridium phylicae* (CPC 19962 and CPC 19965) were selected as the outgroup taxa. The newly generated sequences are shown in red.

**Figure 2 jof-09-01096-f002:**
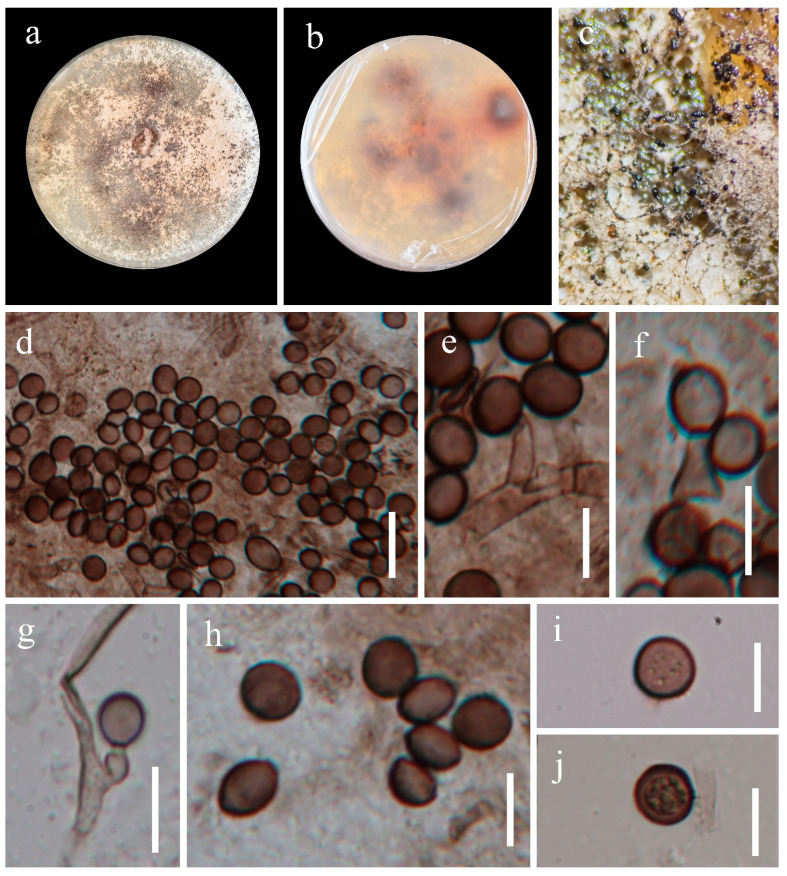
*Apiospora aseptata* (HKAS 129875, holotype). (**a**,**b**) Cultures on PDA from above and below; (**c**) colonies on PDA; (**d**–**g**) conidiogenous cells with conidia; (**h**–**j**) conidia. Scale bars: (**d**) 20 μm; (**e**–**j**) 10 μm.

**Figure 3 jof-09-01096-f003:**
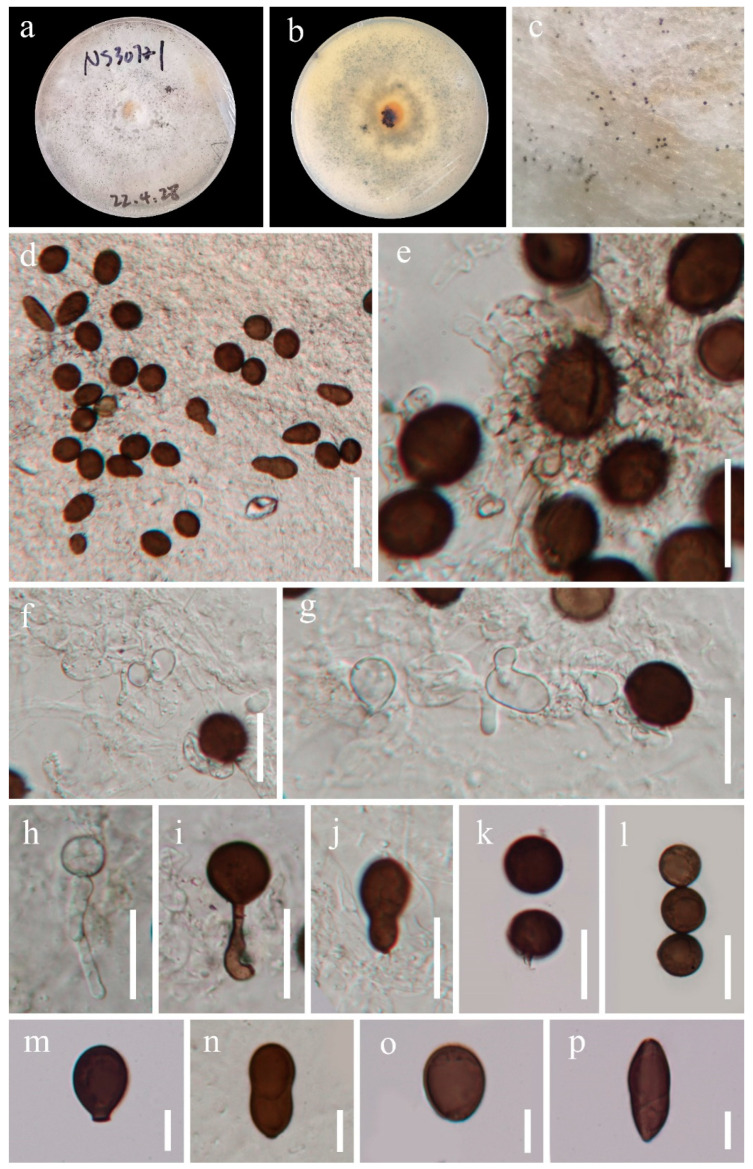
*Apiospora dematiacea* (HKAS 129910, Holotype). (**a**,**b**) Cultures on PDA from above and below; (**c**) colonies on PDA; (**d**–**i**) conidiogenous cells with conidia; (**j**–**p**) conidia. Scale bars: (**d**) 50 μm; (**e**–**l**) 20 μm; (**m**–**p**) 10 μm.

**Figure 4 jof-09-01096-f004:**
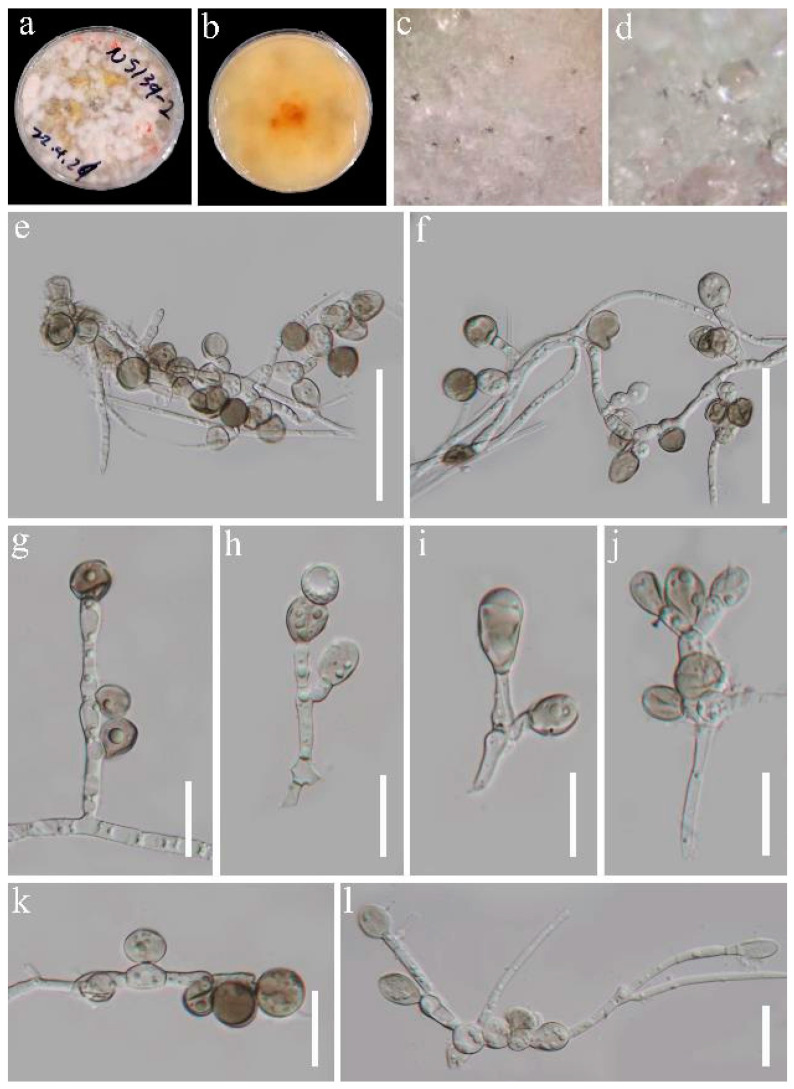
*Apiospora dicranopteridis* (HKAS 129877, holotype). (**a**,**b**) Cultures on PDA from above and below; (**c**,**d**) colonies on PDA; (**d**–**l**) conidiophores, conidiogenous cells with conidia. Scale bars: (**e**,**f**) 50 μm; (**g**–**l**) 20 μm.

**Figure 5 jof-09-01096-f005:**
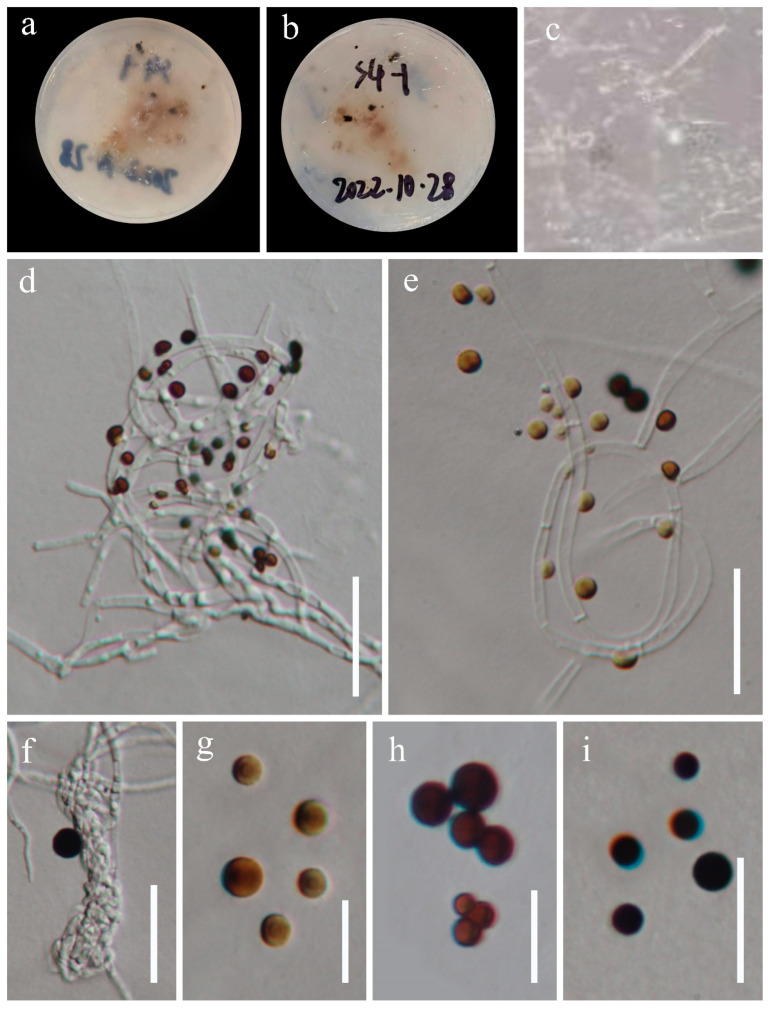
*Apiospora globosa* (HKAS 129921, holotype). (**a**,**b**) Cultures on WA from above and below; (**c**) colonies on WA; (**d**–**f**) conidiophores with conidia; (**g**–**i**) conidia. Scale bars: (**d**–**f**) 20 μm; (**g**–**i**) 10 μm.

**Table 1 jof-09-01096-t001:** *Apiospora* endophytic isolates were used in this study.

Taxon	Strain Code	Specimen	Status	Host	Tissues	Substrate (Sproluration)
*A*. *aseptata*	KUNCC 23-14169	HKAS 129875	H	*D*. *pedata*	Leaf	PDA
*A*. *dematiacea*	KUNCC 23-14202	HKAS 129910	H	*D*. *ampla*	Stem	PDA
*A*. *dicranopteris*	GZCC 23-0708	HKAS 129898	P	*D*. *ampla*	Rhizome	PDA
*A*. *dicranopteris*	GZCC 23-0712	HKAS 129895	P	*D*. *pedata*	Leaf	PDA
*A*. *dicranopteris*	KUNCC23-14177	GZAAS 23-0780	P	*D*. *pedata*	Root	PDA, WA
*A*. *dicranopteris*	KUNCC23-14171	HKAS 129877	H	*D*. *pedata*	Stem	PDA
*A*. *globosa*	KUNCC 23-14210	HKAS 129921	H	*D*. *linearis*	Stem	WA

Note: status: H denotes holotype; P denotes paratype.

**Table 2 jof-09-01096-t002:** Taxa used in this study and their GenBank accession numbers.

Taxa Names	Strains	GenBank Accessions			
		LSU	ITS	TEF1-α	TUB2
*Apiospora acutiapicum*	KUMCC 20-0210 T	MT946339	MT946343	MT947360	MT947366
*Apiospora agari*	KUC21333 T	MH498440	MH498520	MH544663	MH498478
*Apiospora aquaticum*	S 642 T	MK835806	MK828608	N/A	N/A
*Apiospora arctoscopi*	KUC21331 T	MH498449	MH498529	MN868918	MH498487
*Apiospora arundinis*	CBS 449 92	KF144931	KF144887	KF145019	KF144977
*Apiospora arundinis*	GZCC 20-0116 T	MW478899	MW481720	MW522952	MW522968
*Apiospora aurea*	CBS 244.83 T	KF144935	AB220251	KF145023	KF144981
*Apiospora balearica*	CBS 145129 T	MK014836	MK014869	N/A	MK017975
*Apiospora bambusae*	ALV17304	MK014841	MK014874	MK017951	MK017980
*Apiospora bambusicola*	MFLUCC 20-0144 T	MW173087	MW173030	MW183262	N/A
*Apiospora biserialis*	CGMCC 3.20135 T	MW478885	MW481708	MW522938	MW522955
*Apiospora camelliae sinensis*	LC 5007 T	KY494780	KY494704	KY705103	KY705173
*Apiospora chiangraiense*	MFLUCC 21-0053 T	MZ542524	MZ542520	N/A	MZ546409
*Apiospora chromolaenae*	MFLUCC 17-1505 T	MT214436	MT214342	N/A	N/A
*Apiospora cordylinae*	GUCC 10027 T	N/A	MT040106	MT040127	MT040148
*Apiospora cyclobalanopsidis*	CGMCC 3.20136 T	MW478892	MW481713	MW522945	MW522962
*Apiospora descalsii*	CBS 145130 T	MK014837	MK014870		MK017976
*Apiospora dichotomanthi*	LC 4950 T	KY494773	KY494697	KY705096	KY705167
*Apiospora dongyingensis*	SAUCC 0302 T	OP572424	OP563375	OP573264	OP573270
*Apiospora esporlensis*	CBS 145136 T	MK014845	MK014878		MK017983
*Apiospora euphorbiae*	IMI 285638b	AB220335	AB220241	NA	AB220288
*Apiospora fermenti*	KUC21289 T	MF615213	MF615226	MH544667	MF615231
*Apiospora gaoyouensis*	CFCC 52301 T	N/A	MH197124	MH236793	MH236789
*Apiospora garethjonesii*	KUMCC 16-0202 T	KY356091	KY356086	N/A	N/A
*Apiospora gelatinosa*	KHAS 11962 T	MW478888	MW481706	MW522941	MW522958
*Apiospora guiyangensis*	HKAS 102403 T	MW240577	MW240647	N/A	MW775604
*Apiospora guizhouensis*	LC 5322 T	KY494785	KY494709	KY705108	KY705178
*Apiospora hainanensis*	SAUCC 1681 T	OP572422	OP563373	OP573262	OP573268
*Apiospora hispanica*	IMI 326877 T	AB220336	AB220242	N/A	AB220289
*Apiospora hydei*	CBS 114990 T	KF144936	KF144890	KF145024	KF144982
*Apiospora hydei*	LC7103	KY494791	KY494715	KY705114	KY705183
*Apiospora hydei*	LC7105	KY494793	KY494717	KY705116	KY705185
*Apiospora hydei*	SICAUCC 22-0032	ON185553	ON183998	ON221312	ON221313
*Apiospora hyphopodii*	MFLUCC 15-0003 T	N/A	KR069110	N/A	N/A
*Apiospora hyphopodii*	KUMCC 16-0201	KY356093	KY356088	N/A	N/A
*Apiospora hysterina*	ICPM 6889 T	MK014841	MK014874	MK017951	MK017980
*Apiospora iberica*	CBS 145137 T	MK014846	MK014879	N/A	MK017984
*Apiospora intestini*	CBS 135835 T	KR149063	KR011352	KR011351	KR011350
*Apiospora italica*	CBS 145138 T	MK014847	MK014880	MK017956	MK017985
*Apiospora jatrophae*	AMH 9557 T	N/A	JQ246355	N/A	N/A
*Apiospora jiangxiensis*	LC 4577 T	KY494769	KY494693	KY705092	KY705163
*Apiospora kogelbergensis*	CBS 113333 K	KF144938	KF144892	KF145026	KF144984
*Apiospora koreana*	KUC21332 T	MH498444	MH498524	MH544664	MH498482
*Apiospora locuta pollinis*	LC 11683 T	N/A	MF939595	MF939616	MF939622
*Apiospora longistroma*	MFLUCC 11-0481 T	KU863129	KU940141	N/A	N/A
*Apiospora malaysiana*	CBS 102053T	KF144942	KF144896	KF145030	KF144988
*Apiospora marianiae*	AP18219 T	ON692422	ON692406	N/A	ON677186
*Apiospora marii*	CBS 497.90 T	KF144947	AB220252	KF145035	KF144993
*Apiospora marina*	KUC21328 T	MH498458	MH498538	MH544669	MH498496
*Apiospora mediterranea*	IMI 326875	AB220337	AB220243	N/A	AB220290
*Apiospora minutispora*	17E-042 T	N/A	LC517882	LC518889	LC518888
*Apiospora montagnei*	AP301120 T	ON692424	ON692408	ON677182	ON677188
*Apiospora mori*	MFLU 18-2514 T	MW114393	MW114313	N/A	N/A
*Apiospora multiloculata*	MFLUCC 21-0023 T	OL873138	OL873137	N/A	OL874718
*Apiospora mytilomorpha*	DAOM 214595 T	N/A	KY494685	N/A	N/A
*Apiospora neobambusae*	LC 7106 T	KY494794	KY494718	KY806204	KY705186
*Apiospora neochinense*	CFCC 53036 T	N/A	MK819291	MK818545	MK818547
*Apiospora neogarethjonesii*	HKAS 102408 T	MK070898	MK070897	N/A	N/A
*Apiospora neosubglobosa*	JHB 006	KY356094	KY356089	N/A	N/A
*Apiospora neosubglobosa*	KUMCC 16-0203 T	KY356095	KY356090	N/A	N/A
*Apiospora obovata*	LC 4940 T	KY494772	KY494696	KY705095	KY705166
*Apiospora ovata*	CBS 115042 T	KF144950	KF144903	KF145037	KF144995
*Apiospora paraphaeosperma*	MFLUCC 13-0644 T	KX822124	KX822128	N/A	N/A
*Apiospora phragmitis*	CPC 18900	KF144956	KF144909	KF145043	KF145001
*Apiospora phyllostachydis*	MFLUCC 18-1101 T	MH368077	MK351842	MK340918	MK291949
*Apiospora piptatheri*	CBS 145149 T	MK014860	MK014893	N/A	N/A
*Apiospora pseudomarii*	GUCC 10228 T	N/A	MT040124	MT040145	MT040166
*Apiospora pseudoparenchymatica*	LC7234 T	KY494819	KY494743	KY705139	KY705211
*Apiospora pseudorasikravindrae*	KUMCC 20-0208 T	N/A	MT946344	MT947361	MT947367
*Apiospora pseudosinensis*	CPC 21546 T	KF144957	KF144910	KF145044	N/A
*Apiospora pseudospegazzinii*	CBS 102052 T	KF144958	KF144911	KF145045	KF145002
*Apiospora pterosperma*	CPC 20193 T	KF144960	KF144913	KF145046	KF145004
*Apiospora pusillisperma*	KUC21321 T	MH498453	MH498533	MN868930	MH498491
*Apiospora qinlingensis*	CFCC 52303 T	N/A	MH197120	MH236795	MH236791
*Apiospora rasikravindrae*	NFCCI 2144	N/A	JF326454	N/A	N/A
*Apiospora rasikravindrae*	LC5449	KY494789	KY494713	KY705112	KY705182
*Apiospora sacchari*	CBS 372.67	KF144964	KF144918	KF145049	KF145007
*Apiospora saccharicola*	CBS 831.71	KF144969	KF144922	KF145054	KF145012
*Apiospora sargassi*	KUC21228 T	KT207696	KT207746	MH544677	KT207644
*Apiospora sasae*	CBS 146808 T	MW883797	MW883402	N/A	MW890120
*Apiospora septata*	CGMCC 3.20134 T	MW478890	MW481711	MW522943	MW522960
*Apiospora serenensis*	IMI 326869 T	AB220344	AB220250	N/A	AB220297
*Apiospora setariae*	MT492005	N/A	MT492005	MW118457	MT497467
*Apiospora setostroma*	KUMCC 19-0217	MN528011	MN528012	MN527357	N/A
*Apiospora sichuanensis*	HKAS 107008 T	MW240578	MW240648	N/A	MW775605
*Apiospora sorghi*	URM 93000 T	N/A	MK371706	N/A	MK348526
*Apiospora* sp.	SAUCC 1429	OQ615287	OQ592558	N/A	N/A
*Apiospora* sp.	SAUCC 1430	OQ615286	OQ592557	N/A	N/A
*Apiospora sphaerosperma*	CBS114314	KF144951	KF144904	KF145038	KF144996
*Apiospora stipae*	CBS 146804 T	MW883798	MW883403	MW890082	MW890121
*Apiospora subglobosa*	MFLUCC 11-0397 T	KR069113	KR069112	N/A	N/A
*Apiospora subrosea*	LC 7292 T	KY494828	KY494752	KY705148	KY705220
*Apiospora taeanensis*	KUC21322T	N/A	MH498515	MH544662	MH498473
*Apiospora thailandica*	MFLUCC 15-0202 T	KU863133	KU940145	N/A	N/A
*Apiospora tropica*	MFLUCC 21 0056 T	OK491653	OK491657	N/A	N/A
*Apiospora vietnamensis*	IMI 99670 T	KX986111	KX986096	N/A	KY019466
*Apiospora xenocordella*	CBS 478 86 T	KF144970	KF144925	KF145055	KF145013
*Apiospora yunnana*	MFLUCC 15 1002 T	KU863135	KU940147	N/A	N/A
*Arthrinium austriacum*	GZU 345006	MW208860	MW208929	N/A	N/A
*Arthrinium caricicola*	CBS 145127	MK014838	MK014871	N/A	MK017977
*Arthrinium* cf. *sporophleoides*	GZU 345102	MW208866	MW208944	N/A	N/A
*Arthrinium crenatum*	AG 19066 T	MW208861	MW208931	N/A	N/A
*Arthrinium japonicum*	IFO 31098	AB220358	AB220264	N/A	AB220311
*Arthrinium luzulae*	AP7619 3 T	MW208863	MW208937	N/A	N/A
*Arthrinium curvatum var. minus*	CBS 145131	MK014839	MK014872	N/A	MK017978
*Arthrinium morthieri*	GZU 345043	MW208864	MW208938	MW221920	MW221926
*Arthrinium puccinioides*	CBS 145150	MK014861	MK014894	N/A	MK017998
*Arthrinium sphaerospermum*	AP25619	MW208865	MW208943	N/A	N/A
*Arthrinium sporophleum*	CBS 145154	MK014865	MK014898	N/A	MK018001
*Nigrospora aurantiaca*	CGMCC 3.18130 T	KX986098	KX986064	KY019295	KY019465
*Nigrospora bambusae*	CGMCC 3.18327 T	NG 069455	KY385307	KY385313	KY385319
*Nigrospora camelliae sinensis*	CGMCC 3.18125 T	KX986103	KX985986	KY019293	KY019460
*Nigrospora chinensis*	CGMCC 3.18127 T	KX986107	KX986023	KY019422	KY019462
*Nigrospora gorlenkoana*	CBS 480.73 T	KX986109	KX986048	KY019420	KY019456
*Nigrospora guilinensis*	CGMCC 3.18124 T	KX986113	KX985983	KY019292	KY019459
*Nigrospora hainanensis*	CGMCC 3.18129 T	KX986112	KX986091	KY019415	KY019464
*Nigrospora lacticolonia*	CGMCC 3.18123 T	KX986105	KX985978	KY019291	KY019458
*Nigrospora musae*	CBS 319.34 T	KX986110	MH855545	KY019419	KY019455
*Nigrospora oryzae*	LC2693	KX986101	KX985944	KY019299	KY019471
*Nigrospora osmanthi*	CGMCC 3.18126 T	KX986106	KX986010	KY019421	KY019461
*Nigrospora pyriformis*	CGMCC 3.18122 T	KX986100	KX985940	KY019290	KY019457
*Nigrospora rubi*	LC2698 T	KX986102	KX985948	KY019302	KY019475
*Nigrospora saccharicola*	CGMCC 3.19362 T	N/A	MN215788	MN264027	MN329951
*Nigrospora sphaerica*	LC7298	KX986097	KX985937	KY019401	KY019606
*Nigrospora vesicularis*	CGMCC 3.18128 T	KX986099	KX986088	KY019294	KY019463
*Nigrospora zimmermanii*	CBS 290.62 T	KY806276	KY385309	KY385311	KY385317
*Seiridium phylicae*	CPC 19962 T	NG 042759	LT853092	LT853189	LT853239
*Seiridium phylicae*	CPC 19965	KC005809	LT853093	LT853190	LT853240

Note: T denotes type strains; “N/A” indicates no data are available in GenBank.

## Data Availability

All sequences generated in this study were submitted to GenBank.

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
