# Peer review of "Four New Endophytic Apiospora Species Isolated from Three Dicranopteris Species in Guizhou, China"

_jof, 2023, doi:10.3390/jof9111096_

Round 1

Reviewer 1 Report

Comments and Suggestions for Authors

The work is very interesting, even more so if it relates the authors' findings to other initiatives to try to understand the endophytic biota that is still little explored. However, the manuscript is very descriptive, there is no contextualization in the introduction. The authors seem to give the air or style of classic taxonomists, which does not add any additional value to the data. Note that I am not criticizing taxonomy as something derogatory, but the way of writing 50 years ago is not the same as it is today. Therefore, an extensive contextualization is needed, perhaps presenting the area where the samples were collected and the phorophytes from where the isolates were obtained and how this can result in knowledge of local, regional or global diversity. Something along these lines would provide a good background to justify all the work effort.

Furthermore, although the methodology is very well described, I still think it needs to be further explored how the primers were obtained and the refinement of molecular analyzes.

The results are also very descriptive, which is no problem at all, but when we get to the discussions, we get the impression that the authors have given up on continuing to write and that for me is the most important part. The authors need to reflect on the findings and write these reflections in the discussions. As it is, it has no validity whatsoever for the manuscript or for the authors' efforts to search for and gather all this work data.

Therefore, I request an extended review to improve and adapt the manuscript. Then I can reevaluate to check whether the journal was satisfactory and what is expected for modern scientific work with robust and coherent information.

Author Response

Thank you very much for your important comments.

Reviewer 2 Report

Comments and Suggestions for Authors

Dear Аuthors!

The manuscript is devoted to the study of species diversity of endophytic fungi of the Apiospora species, which is a popular trend of taxonomic research. To improve the quality of the manuscript, I propose to make some changes to it.

1. The introduction should be expanded. Please write what effect representatives of the Apiospora species have on their hosts and what biologically active compounds they produce.

2. There is no clear purpose of the study in the Introduction.

3. Table 1 should be transferred to the Results section.

4. Line 66. What do the authors mean by the term “host”?

5. Table 2 is too large and should be deleted. It already exists as additional material.

6. Рlease write about the prospects for your further research in the Discussion section.

Comments on the Quality of English Language

Minor editing of English language required

Author Response

Thank you very much for your valuable suggestion,

Round 2

Reviewer 1 Report

Comments and Suggestions for Authors

The author’ s improve the manuscript or justify their chooses in the manuscript style. The results sound much better in this new version